# Disparities in Access to Thoracic Surgeons among Patients Receiving Lung Lobectomy in the United States

Sean J. Halloran [1], Christine E. Alvarado [2], Anuja L. Sarode [3], Boxiang Jiang [2], Jillian Sinopoli [2], Philip A. Linden [2] and Christopher W. Towe [2,*]

1 Department of Surgery, University of Toledo College of Medicine and Life Sciences, Toledo, OH 43614, USA
2 Division of Thoracic and Esophageal Surgery, Department of Surgery, University Hospitals Cleveland Medical Center, 11100 Euclid Avenue Cleveland, Cleveland, OH 44106, USA
3 UH-RISES: Research in Surgical Outcomes and Effectiveness, Department of Surgery, University Hospitals Cleveland Medical Center, Cleveland, OH 44160, USA
* Correspondence: christopher.towe@uhhospitals.org; Tel.: +1-216-844-0405; Fax: +1-216-844-7597

**Abstract:** Objective: Lung lobectomy is the standard of care for early-stage lung cancer. Studies have suggested improved outcomes associated with lobectomy performed by specialized thoracic surgery providers. We hypothesized that disparities would exist regarding access to thoracic surgeons among patients receiving lung lobectomy for cancer. Methods: The Premier Hospital Database was used to identify adult inpatients receiving lung lobectomy from 2009 to 2019. Patients were categorized as receiving their lobectomy from a thoracic surgeon, cardiovascular surgeon, or general surgeon. Sample-weighted multivariable analysis was performed to identify factors associated with provider type. Results: When adjusted for sampling, 121,711 patients were analyzed, including 71,709 (58.9%) who received lobectomy by a thoracic surgeon, 36,630 (30.1%) by a cardiovascular surgeon, and 13,373 (11.0%) by a general surgeon. Multivariable analysis showed that thoracic surgeon provider type was less likely with Black patients, Medicaid insurance, smaller hospital size, in the western region, and in rural areas. In addition, non-thoracic surgery specialty was less likely to perform minimally-invasive (MIS) lobectomy (cardiovascular OR 0.80, $p < 0.001$, general surgery OR 0.85, $p = 0.003$). Conclusions: In this nationally representative analysis, smaller, rural, non-teaching hospitals, and certain regions of the United States are less likely to receive lobectomy from a thoracic surgeon. Thoracic surgeon specialization is also independently associated with utilization of minimally invasive lobectomy. Combined, there are significant disparities in access to guideline-directed surgical care of patients receiving lung lobectomy.

**Keywords:** thoracic surgery; lung lobectomy; disparities; access to care

## 1. Introduction

Studies across multiple areas of surgery have found a relationship between fellowship-trained surgeons and improved outcomes of complex surgical procedures [1–3]. In patients receiving lung lobectomy for lung cancer, specialty training has been found to be an independent predictor of improved morbidity and mortality [4,5]. Furthermore, multiple studies have shown significant benefits to utilization of minimally invasive surgery (MIS) approaches for lung lobectomy in terms of decreased complications, hospital length of stay, and mortality [6–8]. A nationally representative study by Blasberg et al. found significantly higher utilization of MIS for lobectomy amongst thoracic surgeons in comparison to non-thoracic providers [8]. That study also showed significant geographic variation in practice. Blasberg noted that "VATS adoption appears to have slowed in specific regions of the country, where VATS lobectomy rates remain less than 40%". [8] Regional variation has also been demonstrated in other surgical procedures in the United States [9]. Similar findings have been demonstrated within the field of thoracic surgery. Over 75% of thoracic surgeons are employed through either hospital-based or academic/university-based practices, but

the majority of thoracic procedures done in the community setting are performed by general surgeons [10].

Healthcare disparities have been well-documented within the United States [11–13]. The United States is approaching a critical shortage of surgeons, with resultant decreased patient access to surgical specialists [14]. Using data from the American Board of Thoracic Surgery and US Census Bureau, Moffatt-Bruce et al. expect the number of cardiothoracic procedures to increase by 61% and the caseload of the average surgeon to increase by 121% from 2010 to 2035 [15]. We believe that this trend towards increased cardiothoracic caseload will expose shortages in specialty care [16,17]. Specialty care is particularly important for thoracic oncology, where thoracic surgeons play a critical role in the work-up and evaluation of patients with lung cancer. The Nation Comprehensive Cancer Network recommends that decisions about lung cancer surgery "should be performed by thoracic surgeons" [18]. Despite these recommendations, many patients who receive lung resection are not receiving care by a thoracic surgeon. Regional and demographic factors associated with this disparity in access to thoracic surgeons are unknown.

The purpose of this study was to evaluate historical disparities in access to both thoracic surgeons and minimally invasive approaches to surgery among patients receiving lung lobectomy in the United States. We queried a nationally representative database to look for social, racial, and regional differences that may impact how and by whom patients are receiving lung lobectomy. We hypothesize that disparities will be prevalent regarding access to both thoracic surgeons and the minimally invasive approach among lung lobectomy patients, a treatment modality also endorsed by current treatment guidelines [18].

## 2. Materials and Methods

### 2.1. Data Source

This study used the Premier Healthcare Database to analyze disparities and access to thoracic surgeon specialization among patients receiving lobectomy. The Premier Healthcare Database is a nationally representative database that contains de-identified clinical data from more than a thousand participating hospitals, capturing patient billing records, costs, and coding histories. It is comprised of data from more than one billion patient encounters, which equates to approximately twenty-five percent of all inpatient admissions in the United States. The database is maintained by Premier, Inc. (Washington, DC, USA) and contains hospital admissions (patient demographic characteristics), hospital characteristics, surgeon characteristics, payer information, diagnosis-related groups (DRGs), primary and secondary International Classification of Diseases (ICD) diagnosis and procedure codes, current procedural terminology codes, and resource utilization (hospital length-of-stay and in-hospital mortality).

### 2.2. Patient Selection

The Premier Healthcare Database was queried for all adult inpatients age $\geq 18$ years who received an elective lung lobectomy for lung cancer. Procedure codes and diagnosis codes were determined using ICD-9 and -10 version coding. All adult inpatient admissions between 2009 and 2019 were included. Patients were excluded if provider type was unknown, if discharge status was unknown, if a patient had a non-elective admission type, or if the patient's visit status was not inpatient. Patients were categorized by the provider specialty performing the lobectomy: thoracic surgery, cardiovascular surgery, or general surgery. If patients had multiple provider specialties listed, the most specialized provider category was used (thoracic > cardiovascular > general surgery). For the purposes of this study, provider type was analyzed as thoracic vs. non-thoracic. The Elixhauser comorbidity score was generated from ICD-10 coding to estimate comorbidities using software from the Healthcare Cost and Utilization Project (HCUP).

*2.3. Outcome Measures*

The outcome of interest was whether the lobectomy provider was categorized as a thoracic surgery specialist. The secondary outcome measure was whether the lobectomy procedure was performed MIS (defined as video-assisted thoracoscopy or robotic-assisted thoracoscopy) vs. open.

*2.4. Statistical Analysis*

Survey methodology was used to correct for sampling such that patient level weighting derived from the Premier Healthcare Database was used to estimate a nationally representative sample. Hospital and patient characteristics associated with provider specialization were compared using bivariable analysis. Categorical variables were compared using the survey weight-adjusted Pearson's $\chi^2$ test. Explanatory variables from the bivariable analysis that were significant were included in a survey-weighted multivariable logistic regression analysis to determine whether there were demographic and regional differences among provider specializations performing lobectomy. Lastly, given an a priori assumption that MIS is the preferred approach to surgery for lung cancer, we performed a multivariable analysis of factors associated with MIS that included provider specialization.

Statistical analysis was performed using STATA MP (Version 17.0, Statacorp, College Station, TX, USA). Statistical significance was set at a *p* value $\leq 0.05$. Since all patient-related data in the Premier Healthcare Database is aggregated, de-identified, and HIPAA-compliant, this study was determined to be exempt from institution review board review.

**3. Results**

During the study period, the Premier Healthcare Database included 26,999 patients who met inclusion criteria representing an estimated population size of 121,711 lung lobectomy patients. Among them, 71,709 (58.9%) had their surgery performed by a thoracic surgeon while the remaining 50,003 (41.1%) were performed by non-thoracic providers: 36,630 (30.1%) by a cardiovascular surgeon and 13,373 (11.0%) by a general surgeon. These percentages and values are displayed in Figures 1 and 2. Figure 3 represents trends in lobectomy by provider type during the study period. The proportion of lobectomies performed by thoracic surgeons decreased from 2009 to 2019 (71.5% vs. 54.6%).

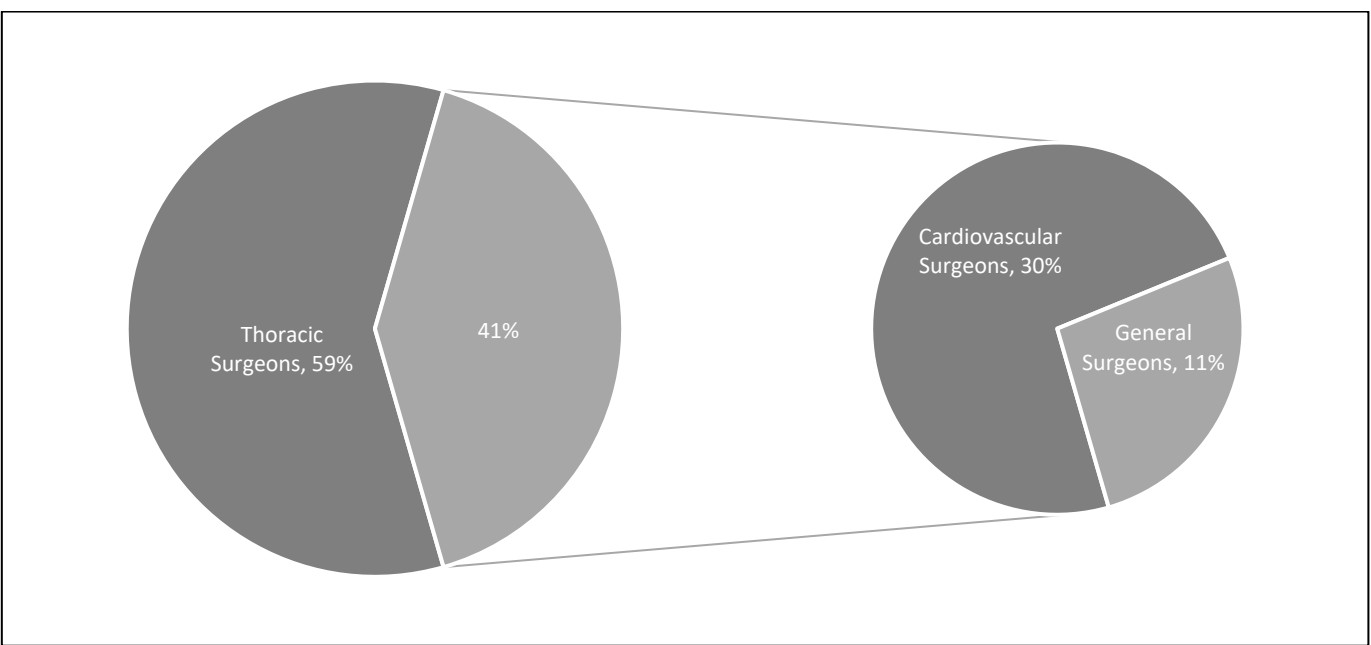

**Figure 1.** Overall percentage of lung lobectomies performed by each provider type in the Premier Healthcare Database from 2009 to 2019.

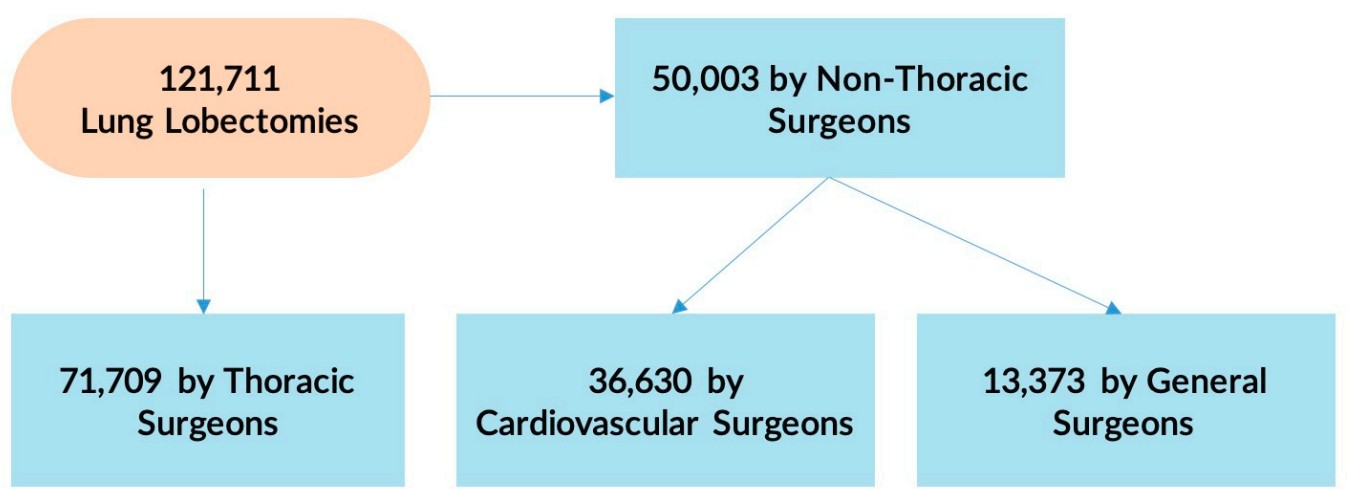

**Figure 2.** Distribution of lung lobectomies performed by each provider type in the Premier Healthcare Database from 2009 to 2019.

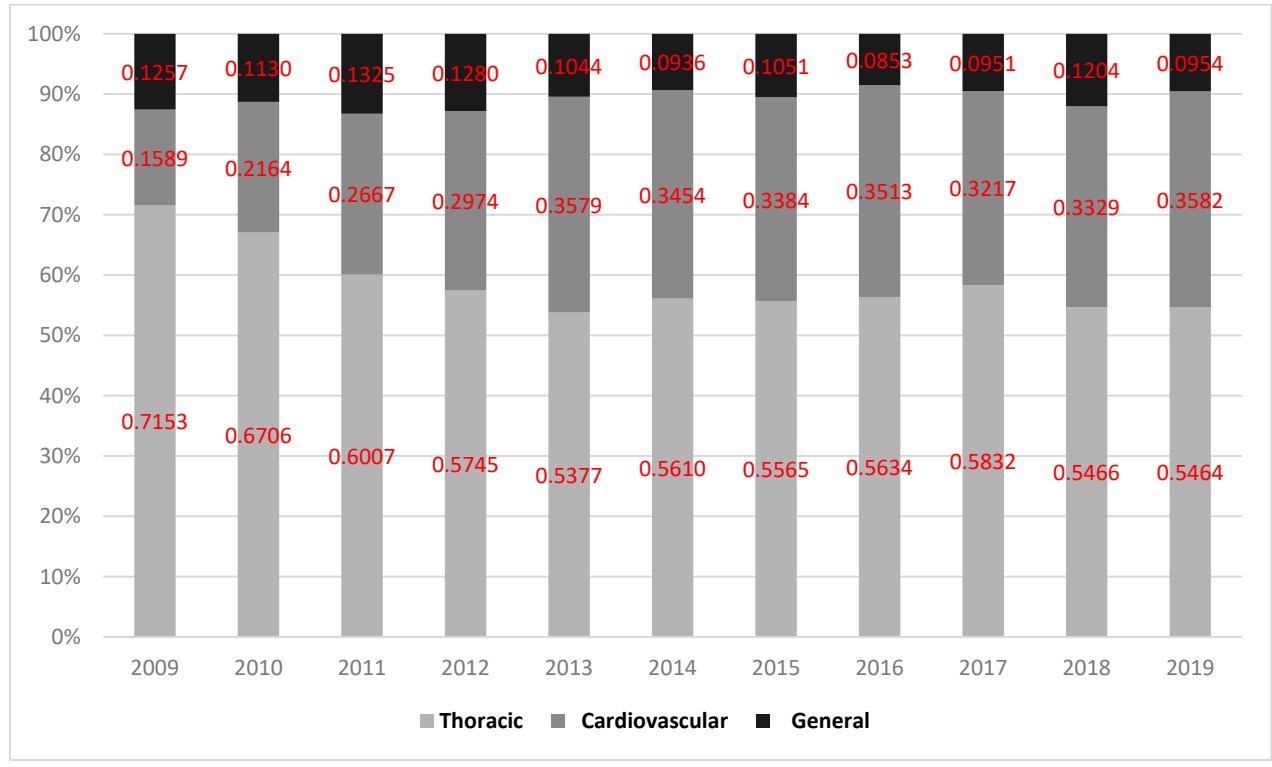

**Figure 3.** Proportion of lung lobectomies performed by each provider type in the Premier Healthcare Database from 2009 to 2019.

Unadjusted analysis revealed several differences between the two groups (Table 1). Patients receiving lobectomy from a thoracic surgeon were more likely to have private insurance (24% vs. 21%, $p < 0.001$), be treated at a hospital with >500 beds (62% vs. 38%, $p < 0.001$), be in an urban setting (61% vs. 40%, $p < 0.001$), and be treated at a teaching hospital (58% vs. 42%, $p < 0.001$).

**Table 1.** Description of cohort of patients in the Premier Healthcare Database (2009 to 2019) stratified by type of provider (thoracic surgeon vs. non-thoracic surgeon) performing the procedure.

| Characteristic | Thoracic Surgeon (n = 71,709) | Non-Thoracic Surgeon (n = 50,003) | *p* Value |
|---|---|---|---|
| Age ≥ 65 (years) | 48,846 (68.1%) | 33,601 (67.2%) | 0.13 |
| Race | | | <0.001 |
|    Asian | 1171 (1.6%) | 867 (1.7%) | |
|    Black | 4874 (6.8%) | 3580 (7.2%) | |
|    White | 41,084 (57.3%) | 26,154 (52.3%) | |
|    Other/unknown | 24,578 (34.3%) | 19,402 (38.8%) | |
| Insurance type | | | <0.001 |
|    Managed | 17,186 (24.0%) | 10,868 (21.7%) | |
|    Medicaid | 3673 (5.1%) | 3159 (6.3%) | |
|    Medicare | 48,396 (67.5%) | 34,140 (68.3%) | |
|    Other (incl charity) | 2455 (3.4%) | 1835 (3.7%) | |
| Hospital size (by # of beds) | | | <0.001 |
|    <300 | 11,884 (16.6%) | 9679 (19.4%) | |
|    300–499 | 25,465 (35.5%) | 19,344 (38.7%) | |
|    ≥500 | 34,359 (47.9%) | 20,980 (42.0%) | |
| Hospital location | | | <0.001 |
|    Rural | 4510 (6.3%) | 6155 (12.3%) | |
|    Urban | 67,199 (93.7%) | 43,848 (87.7%) | |
| Teaching hospital (vs. not) | 39,119 (54.6%) | 27,977 (56.0%) | 0.03 |
| Hospital region | | | <0.001 |
|    Midwest | 22,929 (32.0%) | 13,144 (26.3%) | |
|    Northeast | 9256 (12.9%) | 7300 (14.6%) | |
|    South | 28,234 (39.4%) | 16,705 (33.4%) | |
|    West | 11,290 (15.7%) | 12,853 (25.7%) | |
| Co-morbidities | | | |
|    Congestive heart failure | 3803 (5.3%) | 2608 (5.2%) | 0.10 |
|    Cardiac arrhythmia | 4677 (6.5%) | 3989 (8.0%) | <0.001 |
|    Heart valve disease | 2897 (4.0%) | 2069 (4.1%) | 0.71 |
|    Pulmonary hypertension | 1323 (1.8%) | 1081 (2.2%) | 0.08 |
|    Peripheral vascular disease | 2759 (3.9%) | 2001 (4.0%) | 0.55 |
|    Hypertension, complicated | 5772 (8.1%) | 4473 (9.0%) | 0.01 |
|    Neurologic disorder | 1140 (1.6%) | 947 (1.9%) | 0.07 |
|    Chronic pulmonary disease | 39,968 (55.7%) | 27,995 (56.0%) | 0.70 |
|    Renal failure | 5498 (7.7%) | 3840 (7.7%) | 0.97 |
|    Liver disease | 1025 (1.4%) | 576 (1.2%) | 0.06 |
|    Coagulopathy | 378 (0.5%) | 238 (0.5%) | 0.56 |
|    Obesity | 2639 (3.7%) | 2182 (4.4%) | 0.004 |
|    Weight loss | 2128 (3.0%) | 1917 (3.8%) | <0.001 |
|    Fluid/electrolyte disorders | 13,409 (18.7%) | 9520 (19.0%) | 0.51 |
|    Alcohol abuse | 688 (1.0%) | 567 (1.1%) | 0.15 |
|    Drug abuse | 981 (1.4%) | 851 (1.7%) | 0.04 |

# = Number.

Multivariable analysis (Table 2) showed that thoracic surgeon provider type was less likely in Black patients (odds ratio (OR) 0.84, *p* < 0.001), Medicaid insurance (OR 0.84, *p* = 0.003), smaller hospital size (<300 beds: OR 0.86, *p* < 0.001, 300–499 beds: OR 0.80, *p* < 0.001), western region of the U.S. (OR 0.54, *p* < 0.001), and in rural areas (OR 0.38, *p* < 0.001).

**Table 2.** Multivariable logistic regression analysis of factors associated with patients receiving lung lobectomy from a thoracic surgery provider in the Premier Healthcare Database from 2009 to 2019.

| Characteristic | Odds Ratio | 95% Confidence Interval | *p* Value |
|---|---|---|---|
| Age $\geq$ 65 (years) | 1.16 | 1.079–1.257 | <0.001 |
| Female sex | 1.03 | 0.982–1.083 | 0.22 |
| Race | | | |
|    White | ref | | |
|    Asian | 1.38 | 1.132–1.684 | 0.001 |
|    Black | 0.84 | 0.761–0.925 | <0.001 |
|    Other/unknown | 0.87 | 0.821–0.912 | <0.001 |
| Insurance type | | | |
|    Managed | ref | | |
|    Medicare | 0.82 | 0.758–0.890 | <0.001 |
|    Medicaid | 0.84 | 0.748–0.944 | 0.003 |
|    Other (incl charity) | 0.84 | 0.733–0.970 | 0.02 |
| Hospital size (by # of beds) | | | |
|    $\geq$500 | ref | | |
|    <300 | 0.86 | 0.794–0.930 | <0.001 |
|    300–499 | 0.80 | 0.751–0.847 | <0.001 |
| Hospital region | | | |
|    Northeast | ref | | |
|    Midwest | 1.10 | 1.006–1.194 | 0.04 |
|    South | 1.19 | 1.099–1.295 | <0.001 |
|    West | 0.54 | 0.486–0.590 | <0.001 |
| Hospital location | | | |
|    Urban | ref | | |
|    Rural | 0.38 | 0.349–0.422 | <0.001 |
| Teaching hospital (vs. not) | 1.42 | 1.336–1.504 | <0.001 |

# = Number.

To determine factors associated with the MIS approach to lobectomy, a multivariable analysis was performed, which demonstrated similar disparities (Table 3). MIS was less likely in rural settings (OR 0.75, *p* < 0.001), non-teaching hospitals (OR 0.87, *p* = 0.0001), and in the western region (OR 0.49, *p* < 0.001). A non-thoracic surgeon specialist was also less likely to perform MIS lobectomy (cardiovascular OR 0.80, *p* < 0.001, general surgery OR 0.85, *p* = 0.003). In contrast, non-White patients were more likely to receive MIS lobectomy (Asian OR 5.62, *p* < 0.001, Black OR 1.75, *p* < 0.001).

**Table 3.** Multivariable logistic regression analysis of factors associated with minimally invasive surgical approach among lung lobectomy patients in the Premier Healthcare Database from 2009 to 2019.

| Characteristic | Odds Ratio | 95% Confidence Interval | *p* Value |
|---|---|---|---|
| Age $\geq$ 65 (years) | 1.37 | 1.24–1.51 | <0.001 |
| Female sex | 1.07 | 1.01–1.14 | 0.032 |
| Race | | | |
|    White | ref | | |
|    Asian | 4.53 | 3.67–5.60 | <0.001 |
|    Black | 1.59 | 1.40–1.82 | <0.001 |
|    Other/unknown | 2.98 | 2.78–3.20 | <0.001 |

**Table 3.** *Cont.*

| Characteristic | Odds Ratio | 95% Confidence Interval | *p* Value |
|---|---|---|---|
| Insurance type | | | |
| Managed | ref | | |
| Medicare | 0.85 | 0.77–0.95 | 0.003 |
| Medicaid | 0.88 | 0.76–1.03 | 0.003 |
| Other (incl charity) | 0.66 | 0.53–0.81 | <0.001 |
| Hospital size (by # of beds) | | | |
| ≥500 | ref | | |
| <300 | 0.85 | 0.76–0.94 | 0.001 |
| 300–499 | 0.65 | 0.60–0.71 | <0.001 |
| Surgeon type | | | |
| Thoracic | ref | | |
| Cardiovascular | 0.75 | 0.70–0.81 | <0.001 |
| General | 0.80 | 0.71–0.90 | <0.001 |
| Hospital region | | | |
| Northeast | ref | | |
| Midwest | 0.66 | 0.60–0.74 | <0.001 |
| South | 0.68 | 0.62–0.75 | <0.001 |
| West | 0.51 | 0.45–0.58 | <0.001 |
| Hospital location | | | |
| Urban | ref | | |
| Rural | 0.87 | 0.70–0.91 | 0.001 |
| Teaching hospital (vs. not) | 0.87 | 0.81–0.94 | 0.001 |

\# = Number.

## 4. Discussion

Lung lobectomy is a complex procedure used to treat both benign and malignant lung disease and has been shown to have superior outcomes when performed by a thoracic surgeon [19]. Technological advancements have resulted in the increased adoption of minimally invasive techniques that have demonstrated superior outcomes in comparison to the traditional, open approach [6–8]. This study found multiple social, racial, and regional factors that significantly affected whether a patient receiving a lung lobectomy for cancer would be treated by a thoracic provider. Factors that were associated with a decreased likelihood of receiving care from a thoracic provider included non-White race, treatment in the western U.S., lower socioeconomic status, and rural hospital setting. These findings highlight disparities that exist within our current healthcare system regarding patient access to specialized surgical providers for lung lobectomy.

In addition to these disparities observed for access to thoracic surgeons, similar demographic factors were also associated with a decreased likelihood of MIS utilization. Furthermore, non-thoracic providers were found to have decreased MIS utilization, emphasizing the importance of appropriate access to specialty surgical providers to improve patient outcomes. Bringing awareness to these disparities will help to facilitate discussion, strategies, and hopefully solutions that will increase the accessibility of specialized surgical providers, particularly among these vulnerable patient populations, in order to provide all lung cancer patients with the highest standard of surgical care.

Race has long been recognized as a surrogate for other disparities in the medical field. Byrd et al. demonstrated significant disparities in the prevalence of racial and ethnic minorities that became apparent as early as the preschool years for these groups [20]. They found minority status to have influence on factors such as diet, physical activity, psychological factors, stress, income, and discrimination [20]. These findings have been found to hold true in the field of oncology, as well. Shavers et al. found evidence of racial disparities in receipt of definitive primary therapy, conservative therapy, and adjuvant

therapy for patients with cancer [21]. Our study corroborates these disparities given our finding of non-White patients being less likely to receive care from a thoracic provider.

Interestingly, our study found patients of black race to be more likely to undergo lung lobectomy via a minimally invasive approach; which goes against our original hypothesis. A plausible explanation for this revolves around the geographical distribution of non-white patients in the United States. Non-white patients make up the majority of the population in urban settings [22]. Similarly, the majority of academic medical centers are primarily located in densely populated urban centers. This combination could be the driving force behind increased adoption of minimally invasive lobectomy among non-white patients given that academic medical centers are more likely to utilize minimally invasive approaches [8].

The disparities found in this study have also been observed in other surgical sub-specialties, such as urology and obstetrics and gynecology [23,24]. Boyd et al. found that women of racial minorities who were eligible to receive minimally invasive hysterectomies were significantly more likely to have the procedure done via an open approach and were subject to increased adverse outcomes as a result [23]. In addition, a recently published study among lung and colorectal cancer patients examining the influence of race, insurance status, and rurality found that uninsured status was the largest predictor of receipt of surgery [25]. The gap in access to care is growing larger each year and will only continue to worsen the existing, non-modifiable disparities affecting patients. Increasing both patient access to specialized care through greater outreach and the number of practicing specialized providers are some of the only means to alleviate these disparities.

Previous studies suggest that specialty training improves outcomes in patients under-going lobectomy [4,5]. In addition, numerous studies show fewer complications, shorter length of hospital stay, and improved mortality when lobectomies are performed via a MIS approach [6–8]. In an analysis of Medicare patients, Farjah et al. found that, when adjusting for other patient, hospital, and surgeon factors, specialty training in general thoracic surgery was associated with a significantly decreased risk of death after pulmonary resection for cancer [4]. Based on these findings, referral to specialized thoracic surgeons remains a best practice for the surgical management of lung cancer.

Surgical volume is another key factor to consider in terms of improving lobectomy outcomes [26]. Blasberg et al. found a significant association between MIS utilization and surgeon volume independent of surgeon specialty [8]. In our study, we found that, averaged over the five-year study period, thoracic surgeons had the highest annual rate of lobectomy performance in the country. However, we also found that the proportion of lobectomies performed by thoracic surgeons has decreased over the same five-year period, which may reflect the beginning of a decline in provider specialists.

The findings of this study naturally beg the question of "what can be done to address these disparities?" Given the current fragmented state of the United States healthcare system, there is no simple solution to address these disparities. In the case of thoracic surgery, a plausible solution would be to implement regionalization of specialized thoracic surgery care. Although this approach may seem radical, regionalization of thoracic surgery is not an unprecedented policy. In 2007, Ontario, Canada implemented a policy to regionalize lung cancer surgery to 14 designated hospitals in the province [27]. This policy shift required significant support through government funding, but has resulted in shorter hospital length of stay and decreased mortality amongst certain populations undergoing thoracic procedures [27,28]. Within the United States, the Kaiser Permanente Northern California medical system implemented regionalization of higher complexity thoracic procedures such as lobectomy, bilobectomy, and pneumonectomy to five of the region's 21 hospitals in 2014 [29]. Over a 3-year period, this hospital system demonstrated significant increases in pulmonary resection volume, adoption of a video-assisted thoracoscopic approach, and found regionalization to be independently associated with significant reductions in length of stay and morbidity. This system also demonstrated decreases in 1-year, 3-year, and overall mortality rates following implementation of regionalization [30]. These studies

provide encouraging results to support the regionalization of thoracic care to address the disparities highlighted in our study.

This study proves to be a timely contribution to the literature, given that our data summarizes the current state of access to thoracic surgeons for lung lobectomies up until the onset of the COVID-19 pandemic. Nation-wide lockdown resulted in a temporary halt to all elective procedures, including lung lobectomy, for an extended period of time. The long-term effects of the pandemic on access to care have yet to be properly characterized. However, Nguyen et al. found rebound increases in surgical volumes following staged reopening of their thoracic oncology program in response to significant drops in volume at the height of the pandemic [31]. Future studies are warranted to properly evaluate the impact of the pandemic on access to specialized thoracic providers.

Our study has several limitations. This is a retrospective review of a large national database. As such, the Premier Healthcare Database lacks granular data surrounding clinical staging, neoadjuvant treatment, and intraoperative and surgical data, which therefore did not allow us to compare important oncologic and survival outcomes between the two groups. Given the lack of short- and long-term mortality data in this study, we hope this study may serve as the basis of future studies evaluating the impact of surgeon specialty on both short- and long-term mortality rates for specialized thoracic procedures such as lung lobectomy. Additionally, it is possible that the database lacks granularity in the administrative coding of race. PHD only represents roughly 25% of all admissions in the United States. Thus, it is possible the database does not truly capture our patient population in the most homogenous manner. Another notable limitation of our study is that it may underestimate the disparities identified in this paper. Patients who did not have access to a specialty provider may not have been offered surgery, which is inferior to non-surgical care for early-stage lung cancer patients.

## 5. Conclusions

In this study of a nationally representative database, factors associated with a decreased likelihood of patients receiving care from a thoracic provider for lung lobectomy included non-White race, lower socioeconomic status, western region, and receipt of care in a rural hospital setting. These factors, with the exception of race, were also associated with a decreased likelihood of a MIS approach to lobectomy. Since it has been demonstrated that both thoracic training and MIS result in better outcomes for lung lobectomy, it is necessary that the appropriate steps are taken to address these disparities and provide easier access to thoracic surgeons, providing all patients with the highest standard of care.

**Author Contributions:** Conceptualization, S.J.H., C.E.A. and C.W.T.; data curation, A.L.S.; formal analysis, A.L.S. and C.W.T.; methodology, C.W.T.; writing—original draft, S.J.H., C.E.A. and C.W.T.; writing—review and editing, S.J.H., C.E.A., B.J., J.S., P.A.L. and C.W.T. All authors have read and agreed to the published version of the manuscript.

**Funding:** This research received no external funding.

**Institutional Review Board Statement:** Since all patient-related data in the PHD is aggregated, de-identified, and HIPAA-compliant, this study was determined to be exempt from institution review board review.

**Informed Consent Statement:** Patient consent was waived due to the de-identified and HIPAA-compliant nature of PHD data.

**Data Availability Statement:** The data presented in this study are available on request from the corresponding author.

**Conflicts of Interest:** Towe reports that he is a consultant and recipient of a grant for Zimmer Biomet. He is also a consultant for SigMedical, Atricure, and Medtronic. The funders had no role in the design of the study; in the collection, analysis, or interpretation of data; in the writing of the manuscript; or in the decision to publish the results. The other authors have no disclosures, sources of funding, or financial relationships to declare.

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
