# Peer review of "Disparities in Access to Thoracic Surgeons among Patients Receiving Lung Lobectomy in the United States"

_curroncol, doi:10.3390/curroncol30030213_

Round 1

Reviewer 1 Report

This paper evaluated access to thoracic surgical cancer care. However, I feel that it has several major issues.

The hypothesis that discrepancies in thoracic surgical cancer care may exist by geography, ethnicity, socioeconomic factors, and so on, is reasonable, but I find that very little rationale is provided in the introduction. Such a rationale is necessary to argue how and why these factors would be expected to influence access to thoracic surgery, and what data are required to answer the ensuing questions.

I would suggest that in fact, the available data, and in particular the data on ethnicity are extremely limited. The only racial groups accounted for are black, white, and Asian. What about hispanics, first nations peoples, and so forth? In addition, are all these groups homogenous, or are there ethnic or other subgroups that warrant separate consideration? What about level of schooling and questions of patient education? If we don't have an adequate starting hypothesis, it is difficult to know what kind of information we would need to extract. I understand that such detailed information may not be available in the database that was used, but nevertheless if that is the case it must be acknowledged as a major limitation that severely affects the possibility of reaching any formal conclusions.

A very interesting finding is that non-white patients were in fact MORE likely to receive MIS lobectomy. This finding is contrary to the authors' hypothesis and goes against the other results as well. This is an especially important result since the authors argue that open surgery is associated with a higher risk of adverse events, yet is absent from the discussion. How do the authors account for and interpret these findings?

I would argue that this paper has implications that go well beyond any considerations of race or socio-economic status. Surely the overarching objective here is to allow access to timely and competent surgical care. When affirming that all patients should be provided with "the highest standards of care", are the authors making a distinction between competent care and "highest standards of care"? What would administrative and credentialing bodies, as well as professional associations (cardiovascular surgeons, general surgeons) have to say about this? And to what extent does MIS represent "competent", or "foremost" care, while open surgery represents an adequate alternative?

And if the objective is to provide the foremost possible care to all, what are the solutions, knowing that many of the providers will be non-thoracic surgeons for the foreseeable future? Given the enormous demands of thoracic surgery specialization on trainees, it is completely predictable that most new graduates would indeed elect to join groups already practicing in urban centers and university hospitals. And so it would seem unrealistic to think that thoracic manpower can ever be sufficiently increased across all  areas serving different populations. Perhaps the solutions then rest in focused training of non-thoracic surgeons in the latest techniques and well conceived competency maintenance programs? And-or perhaps organized networks where thoracic specialists would provide non-thoracic surgeons with mentorship and support, in one way or another, ? All of these questions are fundamental and would require an in-depth reflection and an extensive discussion.

Thus, I would suggest that this paper is extremely limited in scope and does not allow one to reach any meaningful conclusions. At most, I think it provides the reader with a snapshot of certain (limited) patterns of thoracic surgical cancer care, and some "food for thought" as it were, but little more. I do not think this paper is acceptable as an original article. Perhaps something like a brief letter to the editor would be more appropriate.

Reviewer 2 Report

The authors presented a fascinating paper describing the essential disparities in access to both thoracic surgeons and MIS approaches among patients receiving lung lobectomy in the United States. In general, the article is interesting and reasonably well-written, but it needs corrections.

Abstract - line 30: the abbreviation MIS should already be explained here, in Abstract.

Introduction Lines 50-59: this passage is challenging to read, and the reader quickly loses interest. Please select the most critical facts and describe them in 2 shorter sentences. Some can be moved to the discussion - the introduction is to be as informative as possible.

Patients Selection and Results (lines 113-120) - I encourage you to present this data on Flow Chart.

Regarding the methodology: would it be possible, for example, that, according to the database, the patient was operated on in a general surgery department, you classified this procedure as "general surgeon," but in fact he was operated on by a thoracic surgeon, for example, because this department includes a small thoracic surgery sub-department? Or the patient was operated on in a cardiac surgery ward, and you classified him as having been operated on by a cardiac surgeon, but in fact there is a thoracic surgeon in this ward who performs this particular set of procedures?

Figure 1. - I suggest adding data labels with percentage values on individual bars of the charts. This would facilitate interpretation.

Why do you think Black patients were less likely to be operated on by a thoracic surgeon?

Discussion - generally well and comprehensively describes the most important results of the paper and confronts them with the available literature. However, it would be worth quoting from the literature how the Covid-19 pandemic could have influenced this situation in the following years. Is there any funding in the literature showing that the availability of thoracic surgery operations during the pandemic has somehow changed depending on race, medical insurance, or type of hospital? And whether the COVID-19 transition impacted complications in thoracic surgery hospitals. For example, this paper (DOI: 10.1007/s11748-022-01871-x) shows that the incidence of postoperative complications did not differ between the groups of patients with and without COVID-19 history operated by thoracic surgeons. I know that your analysis ends in 2019, but is worth posting at least two sentences as one of the possible directions for further research.

Reviewer 3 Report

Dear Editor and Authors,

Thank you for asking me to review this work titled “Disparities in Access to Thoracic Surgeons Among Patients Receiving Lung Lobectomy in the United States” which investigates the access to specialized thoracic surgeons amongst patients receiving lung lobectomy for cancer in the United States. This is a very interesting subject which as a European Surgeon was not so aware that existed - that is that non specialized in thoracic surgery surgeons in the United States are allowed to perform such a complicated and important procedure as is a lobectomy! As the authors very nicely comment and demonstrate this is more prominent in rural areas and smaller hospitals where access to such specialties is more scarce. However, I agree the issue is quite important because those surgeons, are less likely to perform minimally invasive surgery such as VATS or RATS and also are I feel less oncological focused and proficient!!  This of course, as the authors mention and has been reported  has a significant impact on morbidity and mortality of patients.

This is a database extracted study and although it recruited/mined data from a large pool of patients (121,711) is has all the inherent deficiencies of its retrospective nature and limited number of available variables. It is nevertheless adequately designed, within the limitations mentioned, with clear parameters and methodology. One comment if I may, the Premier Healthcare Database (PHD) only covers about 25% of all admissions in the US. Although one assumes this provides an adequate and unbiased/random set of patients let us not forget it is based on merged and amalgamated health insurance databases – as such there is a possible bias that needs to me mentioned. In addition, there is no short term mortality or long term follow up data provided. Again, this maybe due to the shortcomings of the database and the authors have listed it as such!

In conclusion, this is a well written manuscript which is well presented on an interesting if not universally applicable subject – as mentioned earlier this is not the case in Europe/Asia! I am happy to recommend acceptance pending these really minor comments mentioned. Congratulations to the author.

Round 2

Reviewer 1 Report

I appreciate the authors’ diligence in replying to the reviewers’ comments and revising their paper. However, I maintain my initial opinion. Although this paper does raise some questions worthy of further reflection and discussion, I do not think it meets the standard for an original scientific article.

Author Response

We thank the reviewer for their time and efforts in diligently reviewing our manuscript and for the feedback they have provided. We feel the manuscript is in a better place as a result of their reviewing and feedback. Our revisions, attached in our reply to the Academic Editor, are specifically in response to the comments from the academic editor at this stage in the reviewing process. Thank you again for your time and efforts. 

Reviewer 2 Report

No other comments or suggestions.

Author Response

We thank the reviewer for their time and efforts in diligently reviewing our manuscript and for the feedback they have provided. We feel the manuscript is in a better place as a result of their reviewing and feedback. Our revisions, attached in our reply to the academic editor, are specifically in response to the comments from the academic editor at this stage in the reviewing process. Thank you again for your time and efforts.